# Deep Supervised Discrete Hashing

**Qi Li**     **Zhenan Sun**     **Ran He**     **Tieniu Tan**
Center for Research on Intelligent Perception and Computing
National Laboratory of Pattern Recognition
CAS Center for Excellence in Brain Science and Intelligence Technology
Institute of Automation, Chinese Academy of Sciences
{qli,znsun,rhe,tnt}@nlpr.ia.ac.cn

## Abstract

With the rapid growth of image and video data on the web, hashing has been extensively studied for image or video search in recent years. Benefiting from recent advances in deep learning, deep hashing methods have achieved promising results for image retrieval. However, there are some limitations of previous deep hashing methods (e.g., the semantic information is not fully exploited). In this paper, we develop a deep supervised discrete hashing algorithm based on the assumption that the learned binary codes should be ideal for classification. Both the pairwise label information and the classification information are used to learn the hash codes within one stream framework. We constrain the outputs of the last layer to be binary codes directly, which is rarely investigated in deep hashing algorithm. Because of the discrete nature of hash codes, an alternating minimization method is used to optimize the objective function. Experimental results have shown that our method outperforms current state-of-the-art methods on benchmark datasets.

## 1   Introduction

Hashing has attracted much attention in recent years because of the rapid growth of image and video data on the web. It is one of the most popular techniques for image or video search due to its low computational cost and storage efficiency. Generally speaking, hashing is used to encode high dimensional data into a set of binary codes while preserving the similarity of images or videos. Existing hashing methods can be roughly grouped into two categories: data independent methods and data dependent methods.

Data independent methods rely on random projections to construct hash functions. Locality Sensitive Hashing (LSH) [3] is one of the representative methods, which uses random linear projections to map nearby data into similar binary codes. LSH is widely used for large scale image retrieval. In order to generalize LSH to accommodate arbitrary kernel functions, the Kenelized Locality Sensitive Hashing (KLSH) [7] is proposed to deal with high-dimensional kernelized data. Other variants of LSH are also proposed in recent years, such as super-bit LSH [5], non-metric LSH [14]. However, there are some limitations of data independent hashing methods, e.g., it makes no use of training data. The learning efficiency is low, and it requires longer hash codes to attain high accuracy. Due to the limitations of the data independent hashing methods, recent hashing methods try to exploit various machine learning techniques to learn more effective hash function based on a given dataset.

Data dependent methods refer to using training data to learn the hash functions. They can be further categorized into supervised and unsupervised methods. Unsupervised methods retrieve the neighbors under some kinds of distance metrics. Iterative Quantization (ITQ) [4] is one of the representative unsupervised hashing methods, in which the projection matrix is optimized by iterative projection and thresholding according to the given training samples. In order to utilize the semantic labels of data samples, supervised hashing methods are proposed. Supervised Hashing with Kernels (KSH) [13]

is a well-known method of this kind, which learns the hash codes by minimizing the Hamming distances between similar pairs, and at the same time maximizing the Hamming distances between dissimilar pairs. Binary Reconstruction Embedding (BRE) [6] learns the hash functions by explicitly minimizing the reconstruction error between the original distances and the reconstructed distances in Hamming space. Order Preserving Hashing (OPH) [17] learns the hash codes by preserving the supervised ranking list information, which is calculated based on the semantic labels. Supervised Discrete Hashing (SDH) [15] aims to directly optimize the binary hash codes using the discrete cyclic coordinate descend method.

Recently, deep learning based hashing methods have been proposed to simultaneously learn the image representation and hash coding, which have shown superior performance over the traditional hashing methods. Convolutional Neural Network Hashing (CNNH) [20] is one of the early works to incorporate deep neural networks into hash coding, which consists of two stages to learn the image representations and hash codes. One drawback of CNNH is that the learned image representation can not give feedback for learning better hash codes. To overcome the shortcomings of CNNH, Network In Network Hashing (NINH) [8] presents a triplet ranking loss to capture the relative similarities of images. The image representation learning and hash coding can benefit each other within one stage framework. Deep Semantic Ranking Hashing (DSRH) [26] learns the hash functions by preserving semantic similarity between multi-label images. Other ranking-based deep hashing methods have also been proposed in recent years [18, 22]. Besides the triplet ranking based methods, some pairwise label based deep hashing methods are also exploited [9, 27]. A novel and efficient training algorithm inspired by alternating direction method of multipliers (ADMM) is proposed to train very deep neural networks for supervised hashing in [25]. The classification information is used to learn hash codes. [25] relaxes the binary constraint to be continuous, then thresholds the obtained continuous variables to be binary codes.

Although deep learning based methods have achieved great progress in image retrieval, there are some limitations of previous deep hashing methods (e.g., the semantic information is not fully exploited). Recent works try to divide the whole learning process into two streams under the multi-task learning framework [11, 21, 22]. The hash stream is used to learn the hash function, while the classification stream is utilized to mine the semantic information. Although the two stream framework can improve the retrieval performance, the classification stream is only employed to learn the image representations, which does not have a direct impact on the hash function. In this paper, we use CNN to learn the image representation and hash function simultaneously. The last layer of CNN outputs the binary codes directly based on the pairwise label information and the classification information.

The contributions of this work are summarized as follows. 1) The last layer of our method is constrained to output the binary codes directly. The binary codes are learned to preserve the similarity relationship and keep the label consistent simultaneously. To the best of our knowledge, this is the first deep hashing method that uses both pairwise label information and classification information to learn the hash codes under one stream framework. 2) In order to reduce the quantization error, we keep the discrete nature of the hash codes during the optimization process. An alternating minimization method is proposed to optimize the objective function by using the discrete cyclic coordinate descend method. 3) Extensive experiments have shown that our method outperforms current state-of-the-art methods on benchmark datasets for image retrieval, which demonstrates the effectiveness of the proposed method.

## 2 Deep supervised discrete hashing

### 2.1 Problem definition

Given $N$ image samples $X = \{x_i\}_{i=1}^{N} \in \mathbb{R}^{d \times N}$, hash coding is to learn a collection of $K$-bit binary codes $B \in \{-1, 1\}^{K \times N}$, where the $i$-th column $b_i \in \{-1, 1\}^{K}$ denotes the binary codes for the $i$-th sample $x_i$. The binary codes are generated by the hash function $h(\cdot)$, which can be rewritten as $[h_1(\cdot), ..., h_K(\cdot)]$. For image sample $x_i$, its hash codes can be represented as $b_i = h(x_i) = [h_1(x_i), ..., h_K(x_i)]$. Generally speaking, hashing is to learn a hash function to project image samples to a set of binary codes.

## 2.2 Similarity measure

In supervised hashing, the label information is given as $Y = \{y_i\}_{i=1}^N \in \mathbb{R}^{c \times N}$, where $y_i \in \{0,1\}^c$ corresponds to the sample $x_i$, $c$ is the number of categories. Note that one sample may belong to multiple categories. Given the semantic label information, the pairwise label information is derived as: $S = \{s_{ij}\}$, $s_{ij} \in \{0,1\}$, where $s_{ij} = 1$ when $x_i$ and $x_j$ are semantically similar, $s_{ij} = 0$ when $x_i$ and $x_j$ are semantically dissimilar. For two binary codes $b_i$ and $b_j$, the relationship between their Hamming distance $\text{dist}_H(\cdot, \cdot)$ and their inner product $\langle \cdot, \cdot \rangle$ is formulated as follows: $\text{dist}_H(b_i, b_j) = \frac{1}{2}(K - \langle b_i, b_j \rangle)$. If the inner product of two binary codes is small, their Hamming distance will be large, and vice versa. Therefore the inner product of different hash codes can be used to quantify their similarity.

Given the pairwise similarity relationship $S = \{s_{ij}\}$, the Maximum a Posterior (MAP) estimation of hash codes can be represented as:

$$p(B|S) \propto p(S|B) p(B) = \prod_{s_{ij} \in S} p(s_{ij}|B) p(B) \tag{1}$$

where $p(S|B)$ denotes the likelihood function, $p(B)$ is the prior distribution. For each pair of the images, $p(s_{ij}|B)$ is the conditional probability of $s_{ij}$ given their hash codes $B$, which is defined as follows:

$$p(s_{ij}|B) = \begin{cases} \sigma(\Phi_{ij}), & s_{ij} = 1 \\ 1 - \sigma(\Phi_{ij}), & s_{ij} = 0 \end{cases} \tag{2}$$

where $\sigma(x) = 1/(1 + e^{-x})$ is the sigmoid function, $\Phi_{ij} = \frac{1}{2}\langle b_i, b_j \rangle = \frac{1}{2}b_i^T b_j$. From Equation 2 we can see that, the larger the inner product $\langle b_i, b_j \rangle$ is, the larger $p(1|b_i, b_j)$ will be, which implies that $b_i$ and $b_j$ should be classified as similar, and vice versa. Therefore Equation 2 is a reasonable similarity measure for hash codes.

## 2.3 Loss function

In recent years, deep learning based methods have shown their superior performance over the traditional handcrafted features on object detection, image classification, image segmentation, etc. In this section, we take advantage of recent advances in CNN to learn the hash function. In order to have a fair comparison with other deep hashing methods, we choose the CNN-F network architecture [2] as a basic component of our algorithm. This architecture is widely used to learn the hash function in recent works [9, 18]. Specifically, there are two separate CNNs to learn the hash function, which share the same weights. The pairwise samples are used as the input for these two separate CNNs. The CNN model consists of 5 convolutional layers and 2 fully connected layers. The number of neurons in the last fully connected layer is equal to the number of hash codes.

Considering the similarity measure, the following loss function is used to learn the hash codes:

$$J = -\log\ p(S|B) = -\sum_{s_{ij} \in S} \log\ p(s_{ij}|B) = -\sum_{s_{ij} \in S} \left( s_{ij}\Phi_{ij} - \log\left(1 + e^{\Phi_{ij}}\right) \right). \tag{3}$$

Equation 3 is the negative log likelihood function, which makes the Hamming distance of two similar points as small as possible, and at the same time makes the Hamming distance of two dissimilar points as large as possible.

Although pairwise label information is used to learn the hash function in Equation 3, the label information is not fully exploited. Most of the previous works make use of the label information under a two stream multi-task learning framework [21, 22]. The classification stream is used to measure the classification error, while the hash stream is employed to learn the hash function. One basic assumption of our algorithm is that the learned binary codes should be ideal for classification. In order to take advantage of the label information directly, we expect the learned binary codes to be optimal for the jointly learned linear classifier.

We use a simple linear classifier to model the relationship between the learned binary codes and the label information:

$$Y = W^T B, \tag{4}$$

where $W = [w_1, w_{2,...}, w_C]$ is the classifier weight, $Y = [y_1, y_{2,...}, y_N]$ is the ground-truth label vector. The loss function can be calculated as:

$$Q = L\left(Y, W^T B\right) + \lambda \|W\|_F^2 = \sum_{i=1}^N L\left(y_i, W^T b_i\right) + \lambda \|W\|_F^2, \tag{5}$$

where $L(\cdot)$ is the loss function, $\lambda$ is the regularization parameter, $\|\cdot\|_F$ is the Frobenius norm of a matrix. Combining Equation 5 and Equation 3, we have the following formulation:

$$F = J + \mu Q = - \sum_{s_{ij} \in S} \left( s_{ij} \Phi_{ij} - \log \left( 1 + e^{\Phi_{ij}} \right) \right) + \mu \sum_{i=1}^{N} L \left( y_i, W^T b_i \right) + \nu \|W\|_F^2, \qquad (6)$$

where $\mu$ is the trade-off parameters, $\nu = \lambda \mu$. Suppose that we choose the $l_2$ loss for the linear classifier, Equation 6 is rewritten as follows:

$$F = - \sum_{s_{ij} \in S} \left( s_{ij} \Phi_{ij} - \log \left( 1 + e^{\Phi_{ij}} \right) \right) + \mu \sum_{i=1}^{N} \left\| y_i - W^T b_i \right\|_2^2 + \nu \|W\|_F^2, \qquad (7)$$

where $\|\cdot\|_2$ is $l_2$ norm of a vector. The hypothesis for Equation 7 is that the learned binary codes should make the pairwise label likelihood as large as possible, and should be optimal for the jointly learned linear classifier.

## 2.4  Optimization

The minimization of Equation 7 is a discrete optimization problem, which is difficult to optimize directly. There are several ways to solve this problem. (1) In the training stage, the sigmoid or tanh activation function is utilized to replace the ReLU function after the last fully connected layer, and then the continuous outputs are used as a relaxation of the hash codes. In the testing stage, the hash codes are obtained by applying a thresholding function on the continuous outputs. One limitation of this method is that the convergence of the algorithm is slow. Besides, there will be a large quantization error. (2) The sign function is directly applied after the outputs of the last fully connected layer, which constrains the outputs to be binary variables strictly. However, the sign function is non-differentiable, which is difficult to back propagate the gradient of the loss function.

Because of the discrepancy between the Euclidean space and the Hamming space, it would result in suboptimal hash codes if one totally ignores the binary constraints. We emphasize that it is essential to keep the discrete nature of the binary codes. Note that in our formulation, we constrain the outputs of the last layer to be binary codes directly, thus Equation 7 is difficult to optimize directly. Similar to [9, 18, 22], we solve this problem by introducing an auxiliary variable. Then we approximate Equation 7 as:

$$F = - \sum_{s_{ij} \in S} \left( s_{ij} \Psi_{ij} - \log \left( 1 + e^{\Psi_{ij}} \right) \right) + \mu \sum_{i=1}^{N} \left\| y_i - W^T b_i \right\|_2^2 + \nu \|W\|_F^2,$$
$$s.t. \ \ b_i = \mathrm{sgn}(h_i), \ \ h_i \in \mathbb{R}^{K \times 1}, \ \ (i = 1, ..., N), \qquad (8)$$

where $\Psi_{ij} = \frac{1}{2} h_i^T h_j$. $h_i$ $(i = 1, ..., N)$ can be seen as the output of the last fully connected layer, which is represented as:

$$h_i = M^T \Theta \left( x_i; \theta \right) + n, \qquad (9)$$

where $\theta$ denotes the parameters of the previous layers before the last fully connected layer, $M \in \mathbb{R}^{4096 \times K}$ represents the weight matrix, $n \in \mathbb{R}^{K \times 1}$ is the bias term.

According to the Lagrange multipliers method, Equation 8 can be reformulated as:

$$F = - \sum_{s_{ij} \in S} \left( s_{ij} \Psi_{ij} - \log \left( 1 + e^{\Psi_{ij}} \right) \right)$$
$$+ \mu \sum_{i=1}^{N} \left\| y_i - W^T b_i \right\|_2^2 + \nu \|W\|_F^2 + \eta \sum_{i=1}^{N} \left\| b_i - \mathrm{sgn} \left( h_i \right) \right\|_2^2, \qquad (10)$$
$$s.t. \ \ b_i \in \{-1, 1\}^K, \ \ (i = 1, ..., N),$$

where $\eta$ is the Lagrange Multiplier. Equation 10 can be further relaxed as:

$$F = - \sum_{s_{ij} \in S} \left( s_{ij} \Psi_{ij} - \log \left( 1 + e^{\Psi_{ij}} \right) \right)$$
$$+ \mu \sum_{i=1}^{N} \left\| y_i - W^T b_i \right\|_2^2 + \nu \|W\|_F^2 + \eta \sum_{i=1}^{N} \left\| b_i - h_i \right\|_2^2, \qquad (11)$$
$$s.t. \ \ b_i \in \{-1, 1\}^K, \ \ (i = 1, ..., N).$$

The last term actually measures the constraint violation caused by the outputs of the last fully connected layer. If the parameter $\eta$ is set sufficiently large, the constraint violation is penalized severely. Therefore the outputs of the last fully connected layer are forced closer to the binary codes, which are employed for classification directly.

The benefit of introducing an auxiliary variable is that we can decompose Equation 11 into two sub optimization problems, which can be iteratively solved by using the alternating minimization method.

First, when fixing $b_i$, $W$, we have:

$$\frac{\partial F}{\partial h_i} = -\frac{1}{2} \sum_{j:s_{ij}\in S} \left( s_{ij} - \frac{e^{\Psi_{ij}}}{1+e^{\Psi_{ij}}} \right) h_j - \frac{1}{2} \sum_{j:s_{ji}\in S} \left( s_{ji} - \frac{e^{\Psi_{ji}}}{1+e^{\Psi_{ji}}} \right) h_j - 2\eta \left( b_i - h_i \right) \tag{12}$$

Then we update parameters $M$, $n$ and $\Theta$ as follows:

$$\frac{\partial F}{\partial M} = \Theta \left( x_i; \theta \right) \left( \frac{\partial F}{\partial h_i} \right)^T, \quad \frac{\partial F}{\partial n} = \frac{\partial F}{\partial h_i}, \quad \frac{\partial F}{\partial \Theta(x_i;\theta)} = M \frac{\partial F}{\partial h_i}. \tag{13}$$

The gradient will propagate to previous layers by Back Propagation (BP) algorithm.

Second, when fixing $M$, $n$, $\Theta$ and $b_i$, we solve $W$ as:

$$F = \mu \sum_{i=1}^{N} \left\| y_i - W^T b_i \right\|_2^2 + \nu \left\| W \right\|_F^2 . \tag{14}$$

Equation 14 is a least squares problem, which has a closed form solution:

$$W = \left( BB^T + \frac{\nu}{\mu} I \right)^{-1} B^T Y, \tag{15}$$

where $B = \{b_i\}_{i=1}^N \in \{-1,1\}^{K \times N}$, $Y = \{y_i\}_{i=1}^N \in \mathbb{R}^{C \times N}$.

Finally, when fixing $M$, $n$, $\Theta$ and $W$, Equation 11 becomes:

$$F = \mu \sum_{i=1}^{N} \left\| y_i - W^T b_i \right\|_2^2 + \eta \sum_{i=1}^{N} \left\| b_i - h_i \right\|_2^2,$$
$$s.t. \quad b_i \in \{-1,1\}^K, (i = 1, ..., N). \tag{16}$$

In this paper, we use the discrete cyclic coordinate descend method to iteratively solve $B$ row by row:

$$\min_B \left\| W^T B \right\|^2 - 2 \operatorname{Tr}(P), \quad s.t. \ B \in \{-1,1\}^{K \times N}, \tag{17}$$

where $P = WY + \frac{\eta}{\mu} H$. Let $x^T$ be the $k^{th}$ $(k = 1, ..., K)$ row of $B$, $B_1$ be the matrix of $B$ excluding $x^T$, $p^T$ be the $k^{th}$ column of matrix $P$, $P_1$ be the matrix of $P$ excluding $p$, $w^T$ be the $k^{th}$ column of matrix $W$, $W_1$ be the matrix of $W$ excluding $w$, then we can derive:

$$x = \operatorname{sgn} \left( p - B_1^T W_1 w \right) . \tag{18}$$

It is easy to see that each bit of the hash codes is computed based on the pre-learned $K - 1$ bits $B_1$. We iteratively update each bit until the algorithm converges.

## 3 Experiments

### 3.1 Experimental settings

We conduct extensive experiments on two public benchmark datasets: CIFAR-10 and NUS-WIDE. CIFAR-10 is a dataset containing 60,000 color images in 10 classes, and each class contains 6,000 images with a resolution of 32x32. Different from CIFAR-10, NUS-WIDE is a public multi-label image dataset. There are 269,648 color images in total with 5,018 unique tags. Each image is annotated with one or multiple class labels from the 5,018 tags. Similar to [8, 12, 20, 24], we use a subset of 195,834 images which are associated with the 21 most frequent concepts. Each concept consists of at least 5,000 color images in this dataset.

We follow the previous experimental setting in [8, 9, 18]. In CIFAR-10, we randomly select 100 images per class (1,000 images in total) as the test query set, 500 images per class (5,000 images in

total) as the training set. For NUS-WIDE dataset, we randomly sample 100 images per class (2,100 images in total) as the test query set, 500 images per class (10,500 images in total) as the training set. The similar pairs are constructed according to the image labels: two images will be considered similar if they share at least one common semantic label. Otherwise, they will be considered dissimilar. We also conduct experiments on CIFAR-10 and NUS-WIDE dataset under a different experimental setting. In CIFAR-10, 1,000 images per class (10,000 images in total) are selected as the test query set, the remaining 50,000 images are used as the training set. In NUS-WIDE, 100 images per class (2,100 images in total) are randomly sampled as the test query images, the remaining images (193,734 images in total) are used as the training set.

As for the comparison methods, we roughly divide them into two groups: traditional hashing methods and deep hashing methods. The compared traditional hashing methods consist of unsupervised and supervised methods. Unsupervised hashing methods include SH [19], ITQ [4]. Supervised hashing methods include SPLH [16], KSH [13], FastH [10], LFH [23], and SDH [15]. Both the hand-crafted features and the features extracted by CNN-F network architecture are used as the input for the traditional hashing methods. Similar to previous works, the handcrafted features include a 512-dimensional GIST descriptor to represent images of CIFAR-10 dataset, and a 1134-dimensional feature vector to represent images of NUS-WIDE dataset. The deep hashing methods include DQN [1], DHN [27], CNNH [20], NINH [8], DSRH [26], DSCH [24], DRCSH [24], DPSH [9], DTSH [18] and VDSH [25]. Note that DPSH, DTSH and DSDH are based on the CNN-F network architecture, while DQN, DHN, DSRH are based on AlexNet architecture. Both the CNN-F network architecture and AlexNet architecture consist of five convolutional layers and two fully connected layers. In order to have a fair comparison, most of the results are directly reported from previous works. Following [25], the pre-trained CNN-F model is used to extract CNN features on CIFAR-10, while a 500 dimensional bag-of-words feature vector is used to represent each image on NUS-WIDE for VDSH. Then we re-run the source code provided by the authors to obtain the retrieval performance. The parameters of our algorithm are set based on the standard cross-validation procedure. $\mu$, $\nu$ and $\eta$ in Equation 11 are set to 1, 0.1 and 55, respectively.

Similar to [8], we adopt four widely used evaluation metrics to evaluate the image retrieval quality: Mean Average Precision (MAP) for different number of bits, precision curves within Hamming distance 2, precision curves with different number of top returned samples and precision-recall curves. When computing MAP for NUS-WIDE dataset under the first experimental setting, we only consider the top 5,000 returned neighbors. While we consider the top 50,000 returned neighbors under the second experimental setting.

## 3.2 Empirical analysis

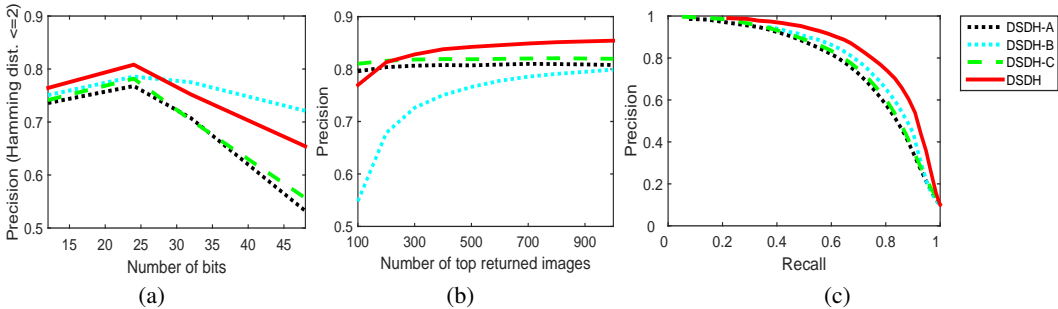

Figure 1: The results of DSDH-A, DSDH-B, DSDH-C and DSDH on CIFAR-10 dataset: (a) precision curves within Hamming radius 2; (b) precision curves with respect to different number of top returned images; (c) precision-recall curves of Hamming ranking with 48 bits.

In order to verify the effectiveness of our method, several variants of our method (DSDH) are also proposed. First, we only consider the pairwise label information while neglecting the linear classification information in Equation 7, which is named DSDH-A (similar to [9]). Then we design a two-stream deep hashing algorithm to learn the hash codes. One stream is designed based on the pairwise label information in Equation 3, and the other stream is constructed based on the classification information. The two streams share the same image representations except for the last

Table 1: MAP for different methods under the first experimental setting. The MAP for NUS-WIDE dataset is calculated based on the top 5,000 returned neighbors. DPSH* denotes re-running the code provided by the authors of DPSH.

| Method | CIFAR-10 | | | | Method | NUS-WIDE | | | |
|--------|----------|----------|----------|----------|--------|----------|----------|----------|----------|
|        | 12 bits  | 24 bits  | 32 bits  | 48 bits  |        | 12 bits  | 24 bits  | 32 bits  | 48 bits  |
| Ours   | **0.740** | **0.786** | **0.801** | **0.820** | Ours   | **0.776** | **0.808** | **0.820** | **0.829** |
| DQN    | 0.554    | 0.558    | 0.564    | 0.580    | DQN    | 0.768    | 0.776    | 0.783    | 0.792    |
| DPSH   | 0.713    | 0.727    | 0.744    | 0.757    | DPSH*  | 0.752    | 0.790    | 0.794    | 0.812    |
| DHN    | 0.555    | 0.594    | 0.603    | 0.621    | DHN    | 0.708    | 0.735    | 0.748    | 0.758    |
| DTSH   | 0.710    | 0.750    | 0.765    | 0.774    | DTSH   | 0.773    | 0.808    | 0.812    | 0.824    |
| NINH   | 0.552    | 0.566    | 0.558    | 0.581    | NINH   | 0.674    | 0.697    | 0.713    | 0.715    |
| CNNH   | 0.439    | 0.511    | 0.509    | 0.522    | CNNH   | 0.611    | 0.618    | 0.625    | 0.608    |
| FastH  | 0.305    | 0.349    | 0.369    | 0.384    | FastH  | 0.621    | 0.650    | 0.665    | 0.687    |
| SDH    | 0.285    | 0.329    | 0.341    | 0.356    | SDH    | 0.568    | 0.600    | 0.608    | 0.637    |
| KSH    | 0.303    | 0.337    | 0.346    | 0.356    | KSH    | 0.556    | 0.572    | 0.581    | 0.588    |
| LFH    | 0.176    | 0.231    | 0.211    | 0.253    | LFH    | 0.571    | 0.568    | 0.568    | 0.585    |
| SPLH   | 0.171    | 0.173    | 0.178    | 0.184    | SPLH   | 0.568    | 0.589    | 0.597    | 0.601    |
| ITQ    | 0.162    | 0.169    | 0.172    | 0.175    | ITQ    | 0.452    | 0.468    | 0.472    | 0.477    |
| SH     | 0.127    | 0.128    | 0.126    | 0.129    | SH     | 0.454    | 0.406    | 0.405    | 0.400    |

fully connected layer. We denote this method as DSDH-B. Besides, we also design another approach directly applying the sign function after the outputs of the last fully connected layer in Equation 7, which is denoted as DSDH-C. The loss function of DSDH-C can be represented as:

$$
\begin{aligned}
F = &-\sum_{s_{ij} \in S} \left(s_{ij}\Psi_{ij} - \log\left(1 + e^{\Psi_{ij}}\right)\right) + \mu \sum_{i=1}^{N} \left\|y_i - W^T h_i\right\|_2^2 \\
&+ \nu \left\|W\right\|_F^2 + \eta \sum_{i=1}^{N} \left\|b_i - \operatorname{sgn}\left(h_i\right)\right\|_2^2, \quad s.t. \ h_i \in R^{K \times 1}, \ (i = 1, ..., N)
\end{aligned}
\tag{19}
$$

Then we use the alternating minimization method to optimize DSDH-C. The results of different methods on CIFAR-10 under the first experimental setting are shown in Figure 1. From Figure 1 we can see that, (1) The performance of DSDH-C is better than DSDH-A. DSDH-B is better than DSDH-A in terms of precision with Hamming radius 2 and precision-recall curves. More information is exploited in DSDH-C than DSDH-A, which demonstrates the classification information is helpful for learning the hash codes. (2) The improvement of DSDH-C over DSDH-A is marginal. The reason is that the classification information in DSDH-C is only used to learn the image representations, which is not fully exploited. Due to violation of the discrete nature of the hash codes, DSDH-C has a large quantization loss. Note that our method further beats DSDH-B and DSDH-C by a large margin.

### 3.3 Results under the first experimental setting

The MAP results of all methods on CIFAR-10 and NUS-WIDE under the first experimental setting are listed in Table 1. From Table 1 we can see that the proposed method substantially outperforms the traditional hashing methods on CIFAR-10 dataset. The MAP result of our method is more than twice as much as SDH, FastH and ITQ. Besides, most of the deep hashing methods perform better than the traditional hashing methods. In particular, DTSH achieves the best performance among all the other methods except DSDH on CIFAR-10 dataset. Compared with DTSH, our method further improves the performance by $3 \sim 7$ percents. These results verify that learning the hash function and classifier within one stream framework can boost the retrieval performance.

The gap between the deep hashing methods and traditional hashing methods is not very huge on NUS-WIDE dataset, which is different from CIFAR-10 dataset. For example, the average MAP result of SDH is 0.603, while the average MAP result of DTSH is 0.804. The proposed method is slightly superior to DTSH in terms of the MAP results on NUS-WIDE dataset. The main reasons are that there exits more categories in NUS-WIDE than CIFAR-10, and each of the image contains multiple labels. Compared with CIFAR-10, there are only 500 images per class for training, which may not be enough for DSDH to learn the multi-label classifier. Thus the second term in Equation 7 plays a limited role to learn a better hash function. In Section 3.4, we will show that our method will achieve

Table 2: MAP for different methods under the second experimental setting. The MAP for NUS-WIDE dataset is calculated based on the top 50,000 returned neighbors. DPSH* denotes re-running the code provided by the authors of DPSH.

| Method | CIFAR-10 | | | | Method | NUS-WIDE | | | |
|---|---|---|---|---|---|---|---|---|---|
| | 16 bits | 24 bits | 32 bits | 48 bits | | 16 bits | 24 bits | 32 bits | 48 bits |
| Ours | **0.935** | **0.940** | **0.939** | **0.939** | Ours | **0.815** | **0.814** | **0.820** | **0.821** |
| DTSH | 0.915 | 0.923 | 0.925 | 0.926 | DTSH | 0.756 | 0.776 | 0.785 | 0.799 |
| DPSH | 0.763 | 0.781 | 0.795 | 0.807 | DPSH | 0.715 | 0.722 | 0.736 | 0.741 |
| VDSH | 0.845 | 0.848 | 0.844 | 0.845 | VDSH | 0.545 | 0.564 | 0.557 | 0.570 |
| DRSCH | 0.615 | 0.622 | 0.629 | 0.631 | DRSCH | 0.618 | 0.622 | 0.623 | 0.628 |
| DSCH | 0.609 | 0.613 | 0.617 | 0.620 | DSCH | 0.592 | 0.597 | 0.611 | 0.609 |
| DSRH | 0.608 | 0.611 | 0.617 | 0.618 | DSRH | 0.609 | 0.618 | 0.621 | 0.631 |
| DPSH* | 0.903 | 0.885 | 0.915 | 0.911 | DPSH* | N/A | | | |

Table 3: MAP for different methods under the first experimental setting. The MAP for NUS-WIDE dataset is calculated based on the top 5,000 returned neighbors.

| Method | CIFAR-10 | | | | NUS-WIDE | | | |
|---|---|---|---|---|---|---|---|---|
| | 12 bits | 24 bits | 32 bits | 48 bits | 12 bits | 24 bits | 32 bits | 48 bits |
| Ours | **0.740** | **0.786** | **0.801** | **0.820** | 0.776 | **0.808** | **0.820** | **0.829** |
| FastH+CNN | 0.553 | 0.607 | 0.619 | 0.636 | 0.779 | 0.807 | 0.816 | 0.825 |
| SDH+CNN | 0.478 | 0.557 | 0.584 | 0.592 | **0.780** | 0.804 | 0.815 | 0.824 |
| KSH+CNN | 0.488 | 0.539 | 0.548 | 0.563 | 0.768 | 0.786 | 0.790 | 0.799 |
| LFH+CNN | 0.208 | 0.242 | 0.266 | 0.339 | 0.695 | 0.734 | 0.739 | 0.759 |
| SPLH+CNN | 0.299 | 0.330 | 0.335 | 0.330 | 0.753 | 0.775 | 0.783 | 0.786 |
| ITQ+CNN | 0.237 | 0.246 | 0.255 | 0.261 | 0.719 | 0.739 | 0.747 | 0.756 |
| SH+CNN | 0.183 | 0.164 | 0.161 | 0.161 | 0.621 | 0.616 | 0.615 | 0.612 |

a better performance than other deep hashing methods with more training images per class for the multi-label dataset.

## 3.4 Results under the second experimental setting

Deep hashing methods usually need many training images to learn the hash function. In this section, we compare with other deep hashing methods under the second experimental setting, which contains more training images. Table 2 lists MAP results for different methods under the second experimental setting. As shown in Table 2, with more training images, most of the deep hashing methods perform better than in Section 3.3. For CIFAR-10 dataset, the average MAP result of DRSCH is 0.624, and the average MAP results of DPSH, DTSH and VDSH are 0.787, 0.922 and 0.846, respectively. The average MAP result of our method is 0.938 on CIFAR-10 dataset. DTSH, DPSH and VDSH have a significant advantage over other deep hashing methods. Our method further outperforms DTSH, DPSH and VDSH by about $2 \sim 3$ percents. For NUS-WIDE dataset, our method still achieves the best performance in terms of MAP. The performance of VDSH on NUS-WIDE dataset drops severely. The possible reason is that VDSH uses the provided bag-of-words features instead of the learned features.

## 3.5 Comparison with traditional hashing methods using deep learned features

In order to have a fair comparison, we also compare with traditional hashing methods using deep learned features extracted by the CNN-F network under the first experimental setting. The MAP results of different methods are listed in Table 3. As shown in Table 3, most of the traditional hashing methods obtain a better retrieval performance using deep learned features. The average MAP results of FastH+CNN and SDH+CNN on CIFAR-10 dataset are 0.604 and 0.553, respectively. And the average MAP result of our method on CIFAR-10 dataset is 0.787, which outperforms the traditional hashing methods with deep learned features. Besides, the proposed algorithm achieves a comparable performance with the best traditional hashing methods on NUS-WIDE dataset under the first experimental setting.

## 4  Conclusion

In this paper, we have proposed a novel deep supervised discrete hashing algorithm. We constrain the outputs of the last layer to be binary codes directly. Both the pairwise label information and the classification information are used for learning the hash codes under one stream framework. Because of the discrete nature of the hash codes, we derive an alternating minimization method to optimize the loss function. Extensive experiments have shown that our method outperforms state-of-the-art methods on benchmark image retrieval datasets.

## 5  Acknowledgements

This work was partially supported by the National Key Research and Development Program of China (Grant No. 2016YFB1001000) and the Natural Science Foundation of China (Grant No. 61622310).

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
