[Reviews · NeurIPS 2017]

Reviewer 1



The paper proposes a deep hashing approach that assumes a setting where there is a training set with class annotations. The model parameters are learned using a loss based on pairwise similarities and class separability under a linear model. The main novelty is in handling the binary embedding directly during learning, without relaxation, using alternating minimization. Results seem better than all the many baselines presented, across two different datasets. There is also a comprehensive analysis of different options. The writing is clear. I'm not an expert in this area, so my main concern is novelty. It seems like the proposed optimization has been proposed before ([9, 17, 21]), so i assume the difference is that in this paper is that there's also the linear classification component of the loss. The overall approach seems also quite complex and potentially very slow to train (how slow?).

Reviewer 2



The paper proposes a supervised hashing algorithm based on a neural network. This network aims to output the binary codes while optimizing the loss for the classification error. To jointly optimize two losses, the proposed method adopts an alternating strategy to minimize the objectives. Experiments on two datasets show good performance compared to other hashing approaches, including the ones use deep learning frameworks. Pros: The paper is written well and easy to follow. Generally, the idea is well-motivated, where the proposed algorithm and optimization strategy are sound and effective. Cons: Some references are missing as shown below, especially that [1] shares similar ideas. It is necessary to discuss [1] and carry out in-depth performance evaluation to clearly demonstrate the merits of this work. [1] Zhang et al., Efficient Training of Very Deep Neural Networks for Supervised Hashing, CVPR16 [2] Liu et al., Deep Supervised Hashing for Fast Image Retrieval, CVPR16 There are missing details for training the network (e.g., hyper-parameters), so it may not be able to reproduce the results accurately. It would be better if the authors will release the code if the paper is accepted. Since the method adopts an alternating optimization approach, it would be better to show the curve of losses and demonstrate the effectiveness of the convergence. The study in Figure 1 is interesting. However, the results and explanations (Ln 229-233) are not consistent. For example, DSDH-B is not always better than DSDH-A. In addition, the reason that DSDH-C is only slightly better than DSDH-A does not sound right (Ln 231-232). It may be the cause of different binarization strategies, and would require more experiments to validate it. Why is DSDH-B better than DSDH in Figure 1(a) when using more bits?

Reviewer 3



The paper propose a new method for learning hash codes using label and similarity data based on deep neural networks. The main idea of the paper is eq(6) where the authors propose a loss function that takes into account both pairwise similarity information and label information. The experiment section seems adequate and examines several aspect of the algo, e.g. importance of adding label information and comparison with state-of-the-art. couple of points: - there is a similar work in K. Lin, H.-F. Yang, J.-H. Hsiao, and C.-S. Chen. Deep learning of binary hash codes for fast image retrieval. It would be interesting to see a comparison both in terms of network structure and performance. - in eq(4) is it possible to use a softmax classifier instead of the linear one?